# Spatiotemporal characters and influence factors of hand, foot and mouth epidemic in Xinjiang, China

Shuman Sun[1], Zhiming Li[1]*, Xijian Hu[1], Ruifang Huang[2]*

**1** College of Mathematics and System Science, Xinjiang University, Urumqi, China, **2** Xinjiang Uygur Autonomous Region Center for Disease Control and Prevention, Urumqi, China

* zmli@xju.edu.cn (ZL); 1664657593@qq.com (RH)

**Data Availability Statement:** Data cannot be shared publicly because authorization is required to access the database on this website. The data of HFM disease are obtained from Center for Disease Control and Prevention of Xinjiang Uygur

## Abstract

Hand, foot and mouth (HFM) disease is a common childhood illness. The paper aims to capture the spatiotemporal characters, and investigate the influence factors of the HFM epidemic in 15 regions of Xinjiang province from 2008 to 2017, China. Descriptive statistical analysis shows that the children aged 0-5 years have a higher HFM incidence, mostly boys. The male-female ratio is 1.5:1. Through the scanning method, we obtain the first cluster high-risk areas. The cluster time is usually from May to August every year. A spatiotemporal model is proposed to analyze the impact of meteorological factors on HFM disease. Comparing with the spatial model, the model is more effective in terms of $R^2$, AIC, deviation, and mean-square error. Among meteorological factors, the number of HFM cases generally increases with the intensity of rainfall. As the temperature increases, there are more HFM patients. Some regions are mostly influenced by wind speed. Further, another spatiotemporal model is introduced to investigate the relationship between HFM disease and socioeconomic factors. The results show that socioeconomic factors have significant influence on the disease. In most areas, the risk of HFM disease tends to rise with the increase of the gross domestic product, the ratios of urban population and tertiary industry. The incidence is closely related to the number of beds and population density in some regions. The higher the ratio of primary school, the lower the number of HFM cases. Based on the above analysis, it is the key measure to prevent and control the spread of the HFM epidemic in high-risk areas, and influence factors should not be ignored.

## Introduction

Hand, foot and mouth (HFM) disease is an infectious disease with the features of high incidence and infectivity, caused by various human enteroviruses [1]. The main symptoms include fever, oral ulcers, maculopapular rashes, or vesicular sores on hands, feet, or mouths [2]. It can also cause aseptic meningitis, respiratory infections, myocarditis, and even the death of some critically ill patients [3]. HFM disease was classified as the class C notifiable infectious disease in China on 2 May 2008. It has attracted extensive attention from the medical and health

Autonomous Region (http://www.xjcdc.com/; contact via 1178843565@qq.com) for researchers who meet the criteria for access to confidential data. The monthly average temperature (Centigrade), precipitation (mm), and wind speed (m/s) can be obtained from China Meteorological Data Service Center (http://data.cma.cn/en). Socioeconomic factors include the population density (100,000 people/10,000 square kilometer), The ratio of urban population (%), the per capita gross domestic product (yuan/people), the ratio of tertiary industry (%), the ratio of primary school (%) and the number of beds (set/1000 people). All annual data are from Xinjiang Statistical Yearbook (https://data.cnki.net/yearbook/Single/N2020040352). The China Meteorological Data Service Center, Xinjiang Statistical Yearbook, and Statistical yearbook of Xinjiang have no special access privileges to these data, but authorization is required to access the database on these websites.

**Funding:** This work was supported by the Natural Science Foundation of Xinjiang Uygur Autonomous Region (XJ2021G020, XJEDU2017M001, 2018Q011), and the National Natural Science Foundation of China (U1703237, 11661076, 12061070). The funders play the important role in study design, the decision to publish, and the preparation of the manuscript.

**Competing interests:** The authors have declared that no competing interests exist.

institutions [4]. As of 2017, the total number of cases has exceeded 67,000, including more than 100 severe cases and 10 deaths in Xinjiang province [5]. Thus, it is very important to analyze the characteristics and influencing factors for preventing the spread of HFM disease in Xinjiang.

Studies show that the spread of HFM epidemic is related to many factors. Wang et al. [6, 7] found that the transmission of HFM disease had significant seasonal characteristic. Li et al. [8] indicated that the meteorological and socioeconomic factors were the main influencing factors. The meteorological factors mainly include temperature, rainfall, and wind speed [9–11]. The socioeconomic factors had the proportion of tertiary industry, the density of children, and the medical facilities [12]. There are considerable works on the study of HFM disease by summary, comparison, or spatiotemporal scanning method. The descriptive statistical analysis was generally used to describe the demographic, geographical, and temporal features [13]. The cluster time and regions can be obtained by the spatiotemporal scanning method [14, 15]. The statistical models were always proposed to analyze the effects between the disease and influencing factors, such as the generalized linear model, generalized additive model, and spatial regression model [16–18]. However, some of them have not considered the influencing factors [15, 19]. Other studies determined the impact of various factors on the disease, but focused on one of time or space [18, 20]. In practice, the impact of meteorological and socioeconomic factors is not only related to space, but also time in the HFM epidemic [21]. Li et al. [8] provided a spatiotemporal mixture model to study the influence of meteorological factors on the disease, ignoring the socioeconomic factor and the interaction among regions. In our work, the geographically and temporally weighted Poisson regression (GTWPR) model is proposed to investigate the influence of meteorological and socioeconomic factors on the HFM epidemic including the interaction between regions.

The rest of this paper is organized as follows. Data sources, spatiotemporal cluster, and spatiotemporal model are introduced in materials and methods. The main results reflect the distributions of HFM disease, and provide the spatiotemporal cluster of 15 regions in Xinjiang from 2008 to 2017. Through the spatiotemporal models, we can get the influence of meteorological and socioeconomic factors on HFM disease. Finally, a brief conclusion is given.

## Materials and methods

Xinjiang province is the largest district in China, including 5 regions (Altay, Tacheng, Kashgar, Aksu, and Khotan), 5 autonomous regions (Ili, Bortala, Changji, Kizilsu, and Bayingholin), and 5 cities (Urumqi, Karamay, Shihezi, Hami, and Turfan). It is located in north latitude 34° 25′ to 49°10′ and east longitude 73°40′ to 96°23′. Fig 1 shows all regions and cities of Xinjiang. It has a temperate continental climate with large seasonal temperature difference ($-50.2° \sim 49.6°$). Because of the unique geographical location, the annual average temperature is 10.6°C and annual average precipitation is 163.3 mm in Xinjiang. Tianshan mountain divides Xinjiang province into two parts: southern and northern areas. The northern has higher precipitation, and the southern has higher temperature.

### Data sources

The data of HFM disease are obtained from Chinese Center for Disease Control and Prevention, and Center for Disease Control and Prevention of Xinjiang Uygur Autonomous Region. The diagnosis of HFM disease conforms to the clinical standards of national ministry of health [22].

There are 34 meteorological stations in Xinjiang province. The monthly average temperature (°C), precipitation (mm), and wind speed (m/s) can be obtained from China

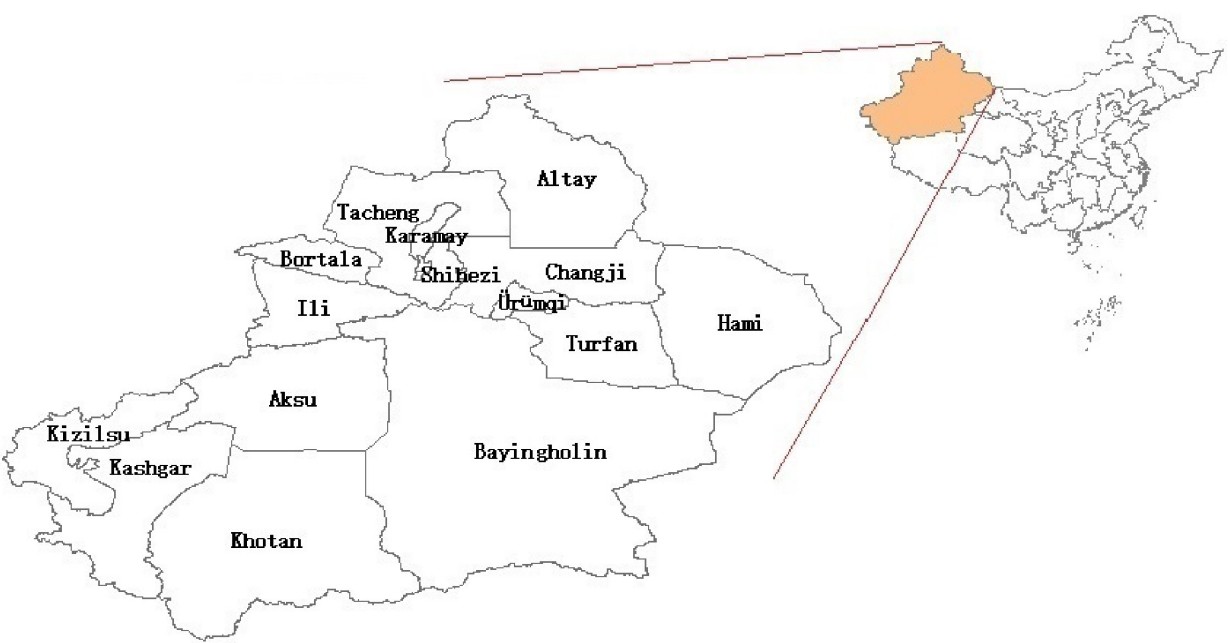

**Fig 1. Map of Xinjiang province, China.** It is generated by software ArcGIS 10.2 from http://eol.jsc.nasa.gov/SearchPhotos/.

Meteorological Data Service Center (http://data.cma.cn/en). Through ArcGIS 10.2, the inverse distance weighting method is used to calculate the average monthly meteorological data in the studied area.

Socioeconomic factors include the population density (100,000 people/10,000 km$^2$), the ratio of urban population (%), the per capita gross domestic product (yuan/people), the ratio of tertiary industry (%), the ratio of primary school (%) and the number of beds (set/1000 people). All annual data are from Xinjiang Statistical Yearbook [23].

## Methods

**Spatiotemporal scanning.** Based on a discrete Poisson model, spatiotemporal scan statistic is defined by a dynamically circular window to detect the possible spatiotemporal cluster. The difference of theoretical and actual cases is calculated by log-likelihood ratio (LLR) statistic [15] as follows

$$\text{LLR} = \log \left( \frac{n}{E(n)} \right)^n \left( \frac{N - n}{N - E(n)} \right)^{N-n} I,$$

where $n$ is the observed number of cases within the scan window, $N$ is the total number of cases, $E(n)$ is the expected number of cases within the scan window, and $I$ is an indicator. It is significant for the relative risk (RR) of scanning window if $p < 0.01$, where $p$-value can be obtained by Monte Carlo randomization. The scan window with the largest LLR value is selected as the first cluster, and other regions with $p < 0.01$ are defined as the second clusters. In this study, the SaTScan software is used to investigate the spatiotemporal clusters of the confirmed cases in the fifteen regions and cities of Xinjiang province.

**Spatiotemporal model.** Fotheringham et al. [24] proposed a geographically weighted Poisson regression (GWPR) model to analyze the spatial non-stationary processes of discrete

data. Suppose that $Y_i(i = 1, \ldots, n)$ are the response variables. The density function is

$$f(y_i; \theta_i, \phi_i) = \exp\left(\frac{y_i\theta_i - b(\theta_i)}{a(\phi_i)} + c(y_i, \phi_i)\right),$$

where $a(\cdot)$, $b(\cdot)$ and $c(\cdot, \cdot)$ are known functions, $\theta_i$ and $\phi_i$ are unknown parameters. Denote $\mu_i = E(Y_i)$, and $g(\mu_i) = \ln(\mu_i)$ is a link function. Let $X_{ij}(i = 1, \ldots, n, j = 1, \ldots, p)$ be explanatory variables and $\beta_j(u_i, v_i)$ be unknown coefficients in the $i$th location $(u_i, v_i)$ and the $j$th variable, respectively. The GWPR model is given by

$$g(\mu_i) \triangleq \eta_i = \beta_0(u_i, v_i) + \sum_{j=1}^{p} \beta_j(u_i, v_i)X_{ij}. \tag{1}$$

In recent years, the spatiotemporal data are often encountered in the geographic area. Comparing with the GWPR model, we propose a geographically and temporally weighted Poisson regression (GTWPR) model. Let $Y_{ik}(i = 1, 2, \ldots, n, k = 1, 2, \ldots, T)$ be the response variable in the $i$th position and the $k$th time. The density function can be defined as follows

$$f(y_{ik}; \theta_{ik}, \phi_{ik}) = \exp\left(\frac{y_{ik}\theta_{ik} - b(\theta_{ik})}{a(\phi_{ik})} + c(y_{ik}, \phi_{ik})\right),$$

where the parameters are similar to the above. Denote $\mu_{ik} = E(Y_{ik})$, and $g(\mu_{ik}) = \ln(\mu_{ik})$. Let $\beta_{jk}(j = 1, 2, \ldots, p, k = 1, 2, \ldots, T)$ be unknown coefficients, and $X_{ijk}(i = 1, 2, \ldots, n_k)$ be the explanatory variables at the $i$th location $(u_{ik}, v_{ik})$ in the $k$th time. The GTWPR model can be established by

$$g(\mu_{ik}) \triangleq \eta_{ik} = \beta_{0k}(u_{ik}, v_{ik}, t_k) + \sum_{j=1}^{p} \beta_{jk}(u_{ik}, v_{ik}, t_k)X_{ijk}. \tag{2}$$

All estimations of these two models can be obtained by weighted least square method and local linear geographical weighted regression. Define a spatiotemporal distance $d_{ik}^{(0)}$ from the fixed point $(u_{00}, v_{00}, t_0)$ to $(u_{ik}, v_{ik}, t_k)$ as

$$d_{ik}^{(0)} = \sqrt{\lambda[(u_{00} - u_{ik})^2 + (v_{00} - v_{ik})^2] + \mu(t_0 - t_k)^2},$$

where $\lambda$ and $\mu$ are space-time balance factors. The Gauss kernel function of these two points can be written by

$$
\begin{aligned}
w_{ik}(u_{00}, v_{00}, t_0) &= \frac{1}{\sqrt{2\pi}}\exp\left\{-\frac{1}{2}\left(\frac{d_{ik}^{(0)}}{h_{ST}}\right)^2\right\} \\
&= \frac{1}{\sqrt{2\pi}}\exp\left\{-\frac{1}{2}\frac{\lambda[(u_{00} - u_{ik})^2 + (v_{00} - v_{ik})^2] + \mu(t_0 - t_k)^2}{h_{ST}^2}\right\} \\
&= \frac{1}{\sqrt{2\pi}}\exp\left\{-\frac{1}{2}\left(\frac{(u_{00} - u_{ik})^2 + (v_{00} - v_{ik})^2}{h_S^2} + \frac{(t_0 - t_k)^2}{\tau h_S^2}\right)\right\},
\end{aligned}
$$

where $h_{ST}$, $h_S$ are the space-time and space bandwidths, $\tau = \lambda/\mu$ is a spatiotemporal factor. The spatiotemporal weight matrix is $W(u_{00}, v_{00}, t_0) = \text{diag}\{w_{ik}(u_{00}, v_{00}, t_0)\}$. Especially, $W(u_{00}, v_{00}, t_0)$ becomes the spatial weight matrix $W(u_{00}, v_{00})$ when time is not considered.

Table 1. Demographic distributions of incidence.

| Category | Type | Year, Incidence(/100,00)(%) | | | | | | | | | | |
|---|---|---|---|---|---|---|---|---|---|---|---|---|
| | | 2008 (N = 4219) | 2009 (N = 6441) | 2010 (N = 7089) | 2011 (N = 5745) | 2012 (N = 9117) | 2013 (N = 5945) | 2014 (N = 7351) | 2015 (N = 7578) | 2016 (N = 10802) | 2017 (N = 3680) | Total (N = 67967) |
| Gender | | | | | | | | | | | | |
| | Male | 2.58(63.8) | 3.67(60.5) | 4.02(61.3) | 3.14(60.1) | 5.03(61.3) | 2.92(59.8) | 3.07(59.7) | 3.60(58.9) | 5.23(59.5) | 1.81(60.0) | 3.50(60.3) |
| | Female | 1.53(36.2) | 2.50(39.5) | 2.64(38.7) | 2.16(39.9) | 3.29(38.7) | 2.03(40.2) | 2.14(40.3) | 2.58(41.1) | 3.63(40.5) | 1.24(40.0) | 2.37(39.7) |
| | Male/femal ratio | 1.8:1 | 1.5:1 | 1.6:1 | 1.5:1 | 1.6:1 | 1.5:1 | 1.5:1 | 1.4:1 | 1.5:1 | 1.5:1 | 1.5:1 |
| Age group | | | | | | | | | | | | |
| | (0, 5) | 22.31 (82.9) | 34.36 (83.6) | 36.82 (81.3) | 29.89 (81.5) | 45.73 (78.6) | 29.43 (77.5) | 36.50 (77.8) | 38.46 (79.5) | 53.79(78.0) | 18.37 (78.2) | 34.56(79.7) |
| | [5, 10) | 4.28(14.1) | 6.03(13.1) | 7.90(15.5) | 6.71(16.3) | 11.85 (18.1) | 7.51(17.6) | 9.96(18.8) | 9.01(16.6) | 14.01(18.1) | 4.68(17.7) | 8.19(16.8) |
| | [10, 15) | 0.60(2.1) | 1.00(2.3) | 0.98(2.1) | 0.56(1.5) | 1.22(2.0) | 1.26(3.2) | 1.08(2.2) | 1.18(2.3) | 1.55(2.2) | 0.64(2.6) | 1.01(2.2) |
| | [15, 20) | 0.07(0.3) | 0.08(0.2) | 0.19(0.5) | 0.08(0.2) | 0.22(0.4) | 0.16(0.5) | 0.11(0.3) | 0.22(0.5) | 0.27(0.4) | 0.06(0.3) | 0.15(0.4) |
| | [20, 120] | 0.02(0.6) | 0.04(0.8) | 0.03(0.6) | 0.02(0.5) | 0.06(0.9) | 0.05(1.2) | 0.04(0.9) | 0.06(1.1) | 0.10(1.3) | 0.03(1.2) | 0.04(0.9) |

* $N$ is the number of cases; $a(b) - a$ is the incidence in the specific gender or age, and $b$ is the proportions of patients with a specific gender or age in the total number of cases.

## Main results

### Distribution characteristics

Demographic, temporal, and geographical distributions are the important basis for understanding the features of a disease. Demographic distribution can explore the risk factors through grouping observational populations. The spread of HFM epidemic can be obtained by temporal distribution in different periods. The geographical distribution provides an important clue to analyze the severity degree of HFM disease in various regions. Matlab and ArcGIS are applied to visualize the distributions of HFM incidence and cases in Xinjiang province from 2008 to 2017.

**Demographic distribution.** Table 1 provides the demographic distributions of HFM disease according to gender and age. There are more than 67,000 reported cases with a male-female ratio 1.5:1. The male has a higher incidence 3.50(1/100,00) than female. All cases are divided into 5 groups by age periods: (0; 5), [5; 10), [10; 15), [15; 20) and [20; 120]. The incidence of children aged 0–5 is higher than others, accounting for 77.5–83.6% of the total cases. Fig 2 shows the gender distribution of the confirmed cases for different ages from 2008 to 2017.

**Temporal distribution.** Based on the monthly reported data, Fig 3 reveals that HFM epidemic has obvious seasonality characteristics from 2008 to 2017. Generally, the disease has one or two peaks every year. The annual peak of cases usually appears in June accounting for 31.29% among 10 years. The second smaller peak is often in October each year.

**Spatial distribution.** The spatial features of incidences are shown in Fig 4. The top three regions are Karamay, Urumqi and Changji with the total incidence 12.73, 9.06 and 6.31(1/100.00), respectively. Moreover, Kizilsu 0.085(1/100.00), Kashgar 0.072(1/100.00) and Khotan 0.686(1/100.00) have relatively lower incidence.

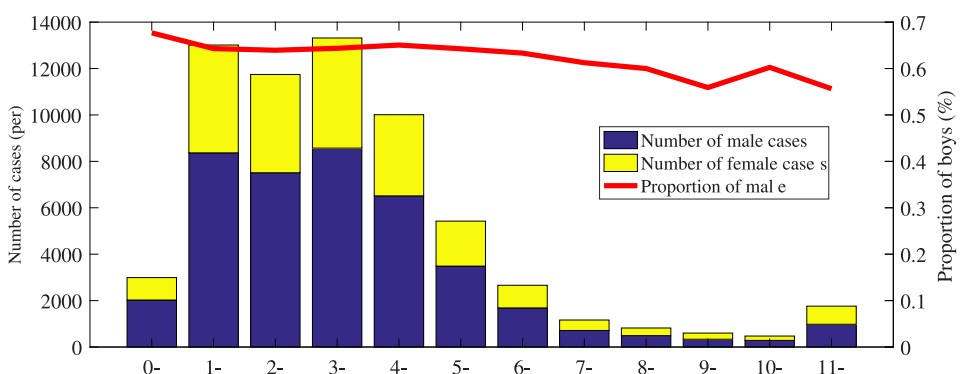

**Fig 2. Gender distribution of the confirmed cases for different ages.**

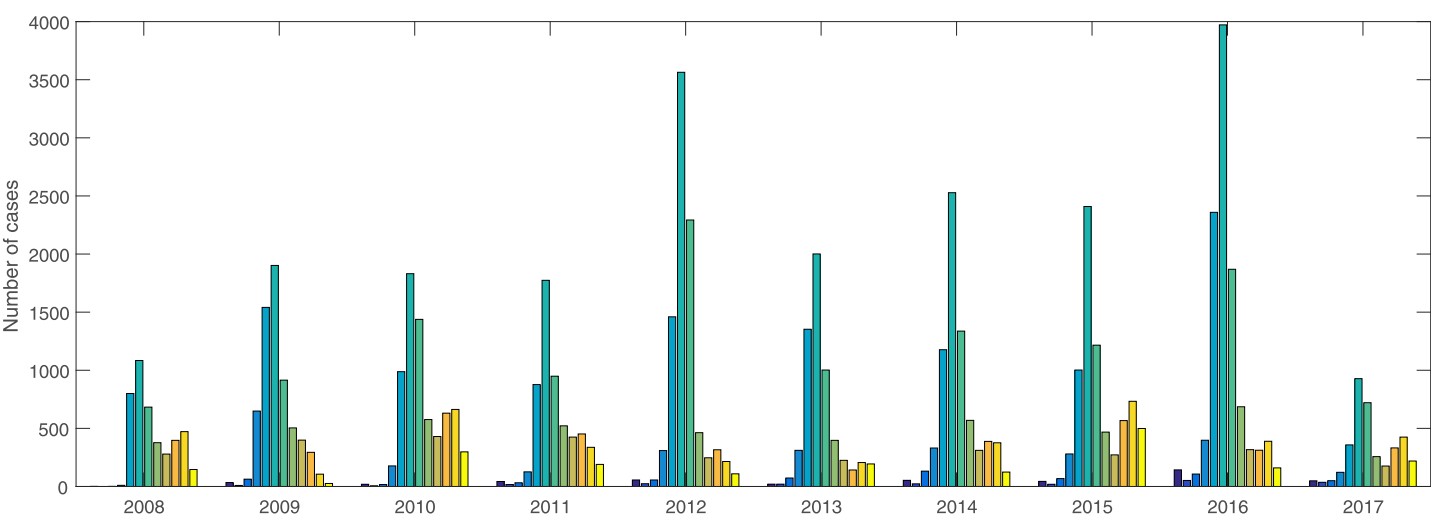

**Fig 3. Seasonal distribution of the confirmed cases.**

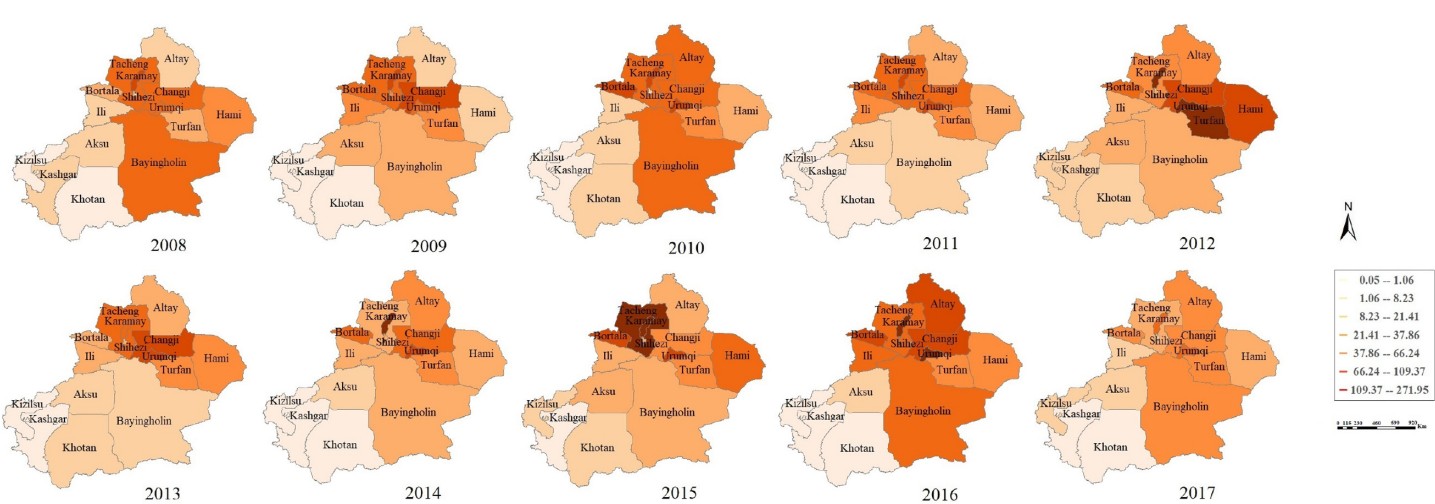

**Fig 4. Spatial distribution of the confirmed incidences.** It is generated by software ArcGIS 10.2 from http://eol.jsc.nasa.gov/SearchPhotos/.

**Table 2. Spatiotemporal clusters of the HFM incidences.**

| Year | Type | Regions and cities | Cluster time | RR | LLR | p-value |
|------|------|--------------------|--------------|-----|-----|---------|
| 2008 | I | Urumqi, Changji, Shihezi Karamay, Bayingholin Turfan, Bortala, Tacheng | May. 10–Nov. 8 | 12.87 | 3093.02 | <0.001 |
|      | II | Aksu, Ili | May. 29–June. 25 | 1.98 | 33.78 | <0.001 |
| 2009 | I | Urumqi, Changji, Shihezi Karamay, Bortala, Tacheng, Ili | Apr. 11–Sept. 29 | 13.11 | 4640.30 | <0.001 |
| 2010 | I | Urumqi, Changji, Shihezi Karamay, Bayingholin, Turfan Bortala, Tacheng, Altay | May. 13–Nov. 10 | 15.87 | 5904.28 | <0.001 |
| 2011 | I | Urumqi, Changji, Shihezi Karamay, Bortala, Tacheng, Ili | Apr. 30–Oct. 6 | 13.02 | 4183.50 | <0.001 |
| 2012 | I | Urumqi, Changji, Shihezi, Karamay, Turfan, Bortala Tacheng, Altay, Hami | May. 1–Aug. 8 | 29.99 | 10357.01 | <0.001 |
| 2013 | I | Urumqi, Shihezi, Changji Karamay, Bortala, Tacheng | Apr. 28–Aug. 12 | 28.23 | 6542.07 | <0.001 |
| 2014 | I | Urumqi, Changji, Shihezi Karamay, Turfan, Bortala Tacheng, Altay | Apr. 19–Aug. 20 | 18.85 | 5812.19 | <0.001 |
| 2015 | I | Urumqi, Changji, Shihezi Karamay, Bayingholin, Turfan Bortala, Altay, Hami | May. 4–Aug. 12 | 15.71 | 4850.71 | <0.001 |
| 2016 | I | Urumqi, Changji, Shihezi Turfan, Bayingholin Karamay, Bortala, Altay | May. 7–Aug. 28 | 30.51 | 10023.10 | <0.001 |
| 2017 | I | Urumqi, Changji, Shihezi Karamay, Bayingholin Turfan, Hami, Altay | May. 28–Nov. 25 | 10.22 | 2455.64 | <0.001 |
| 2008–2017 | I | Urumqi, Changji, Shihezi Karamay, Turfan | Apr. 11. 2012–Sept. 2. 2016 | 6.19 | 19901.47 | <0.001 |
|      | II | Aksu, Ili | Apr. 25. 2009–June. 22. 2009 | 4.45 | 908.85 | <0.001 |

* I—The first cluster area; II—The second cluster area.

## Spatiotemporal cluster

The spatiotemporal scanning method is used to identify the risk regions and time periods of the HFM epidemic in Xinjiang from 2008 to 2017. Table 2 lists the yearly cluster regions and cities, cluster time, RR, LLR and p-value.

The first and second cluster areas are detected for each year, corresponding to the most and more likely to have occurred. We observe that Urumqi, Changji, Shihezi, Karamay and Tacheng always exist in the first cluster regions every year. The second cluster areas are only Aksu and Ili in 2008. Cluster time refers to the clustering time in the specified period. The main cluster time begins in April or May, and ends in August or November each year. Since $RR$ = 29.99, 30.51, and $LLR$ = 10357.01, 10023.10 in 2012 and 2016, it reflects that the spread of HFM epidemic is more serious than that in other years. Moreover, all clustering results have statistically significant because of $p < 0.001$.

Further investigation is conducted by scanning clusters, listed in the last two lines of Table 2. The first clustering areas are Urumqi, Changji, Shihezi, Karamay and Turfan. The second cluster areas are Aksu and Ili. The corresponding cluster times are from April 2012 to September 2016, and from April 2009 to June 2009, respectively.

## Meteorological factors

Sattar et al. [25] and Kramer et al. [26] showed that the low-moderate humidity between temperatures $20°C$ and $33°C$ was more beneficial to the survival and reproduction of viruses on inanimate surfaces. In Xinjiang province, the monthly rainfall is 66.17± 57.40 mm, temperature is 10.38±20.8°C, and wind speed is 19.4± 24.31 m/s. Based on Figs 3 and 5, the HFM epidemic is affected by meteorological factors. Spearman correlation analysis is conducted between lagging meteorological factors and the confirmed cases (Table 3). Lag 0, lag 1, lag 2 and lag 3 represent the factors with lagging zero, one, two and three months, respectively. The results show that rainfall and temperature with lagging zero month and wind speed with lagging one month have a strongly correlated with HFM disease. Therefore, three lagging factors are selected as explanatory variables to establish GWPR and GTWPR models.

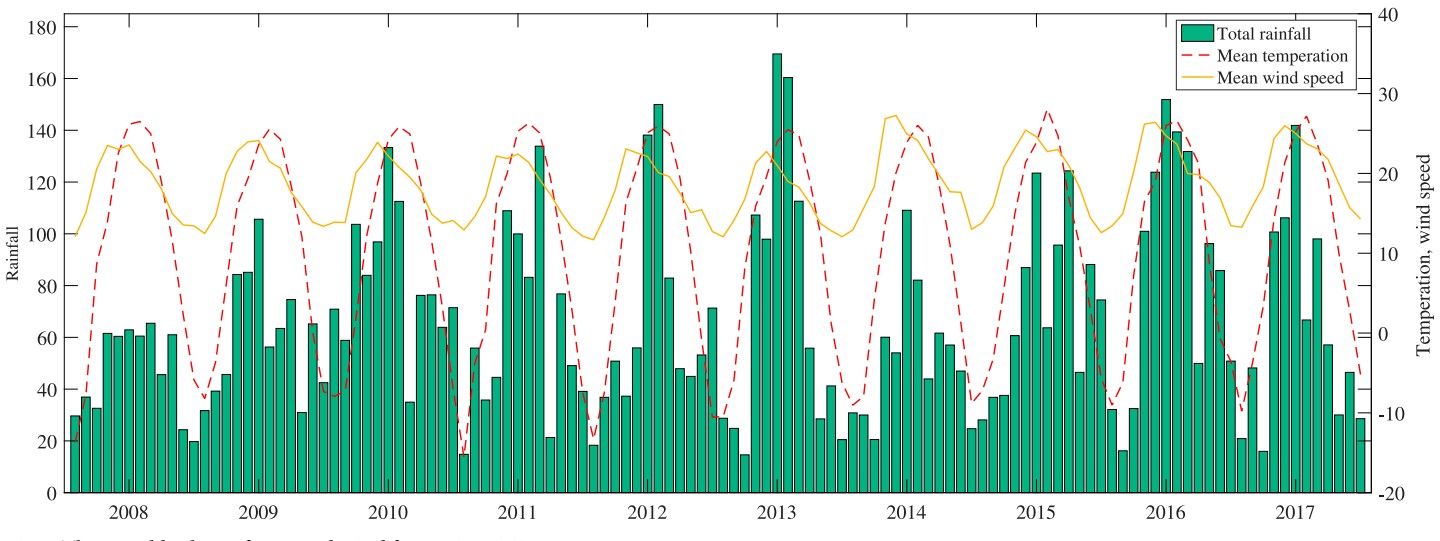

**Fig 5. The monthly chart of meteorological factors in Xinjiang, 2008–2017.**

The meteorological data is first standardized. The multiplex collinear test [27] is determined by the condition number $K = \sqrt{\lambda_{\max}/\lambda_{\min}}$, where $\lambda$ is the eigenvalues of independent variable matrix. Since $K = 1.97 < 15$, there is no significant correlation between meteorological factors. Let $\mu_{ik}$ be the number of cases, and $X_{i1k}$, $X_{i2k}$ and $X_{i3k}$ be the monthly average temperature, rainfall and wind speed in the $k$th month at the $i$th region. The GWPR model is formed by

$$g_1(\mu_{ik}) = \beta_0(u_{ik}, v_{ik}) + \beta_1(u_{ik}, v_{ik})X_{i1k} + \beta_2(u_{ik}, v_{ik})X_{i2k} + \beta_3(u_{ik}, v_{ik})X_{i3k} \tag{3}$$

for $j = 1, 2, 3$, $i = 1, \ldots, n$ and $n = 15$, where $k$ is a fixed constant taken from $\{1, 2, \ldots, m\}$ and $m = 120$. The GTWPR model is established as follows

$$\begin{aligned} g_2(\mu_{ik}) = \quad & \beta_{0k}(u_{ik}, v_{ik}, t_k) + \beta_{1k}(u_{ik}, v_{ik}, t_k)X_{i1k} + \beta_{2k}(u_{ik}, v_{ik}, t_k)X_{i2k} \\ & + \beta_{3k}(u_{ik}, v_{ik}, t_k)X_{i3k}, \quad k = 1, \ldots, m, \end{aligned} \tag{4}$$

where other parameters are the same as above. The estimated coefficients $\hat{\beta}_j(u_{ik}, v_{ik})$ and $\hat{\beta}_{jk}(u_{ik}, v_{ik}, t_k)$ can be obtained by the weighted least square method and local linear geographical weighted regression. The performance of two models are investigated by the coefficient of determination $R^2$, Akaike information criterion (AIC), deviation (D) and mean squared error (MSE) in Table 4. It reveals that $R^2$ increases from 0.633 in the GWPR model to 0.915 in the GTWPR model. The values of AIC, D and MSE in GTWPR model are smaller than GWPR model. Thus, GTWPR model is more suitable and effective to analyze the spatiotemporal HFM disease than GWPR model.

**Table 3. Spearman correlation between lagging factors and the confirmed cases.**

| Meteorological factors | Lag 0 | Lag 1 | Lag 2 | Lag 3 |
|---|---|---|---|---|
| Rainfall (mm) | 0.486 | 0.439 | 0.319 | 0.160 |
| Temperature (˚C) | 0.238 | 0.192 | 0.091 | -0.081 |
| Wind speed (m/s) | 0.187 | 0.223 | 0.199 | 0.110 |

Table 4. The comparison of models (3) and (4).

| Model | $R^2$ | AIC | D | MSE |
|---|---|---|---|---|
| GWPR | 0.633 | 14038.79 | 17010.27 | 0.828 |
| GTWPR | 0.915 | 938.42 | 2367.88 | 0.191 |

According to the GTWPR model, Fig 6 shows the mean values and 95% confidence intervals of coefficient estimations. The mean values are represented by the dots. The 95% confidence interval is represented by the upper and lower lines. There is a significant spatiotemporal non-stationary between the climatic factors and HFM disease. With the increase of the rainfall, the number of HFM cases increases in Shihezi, Urumqi, Changji, Turfan, Hami, and Bayingholin, and decreases in Tacheng, Karamay, Bortala, Ili, Kashgar, and Kizilsu. The rise of temperature leads to an increase in the number of patients in Altay, Tacheng, Karamay, Bortala, Shihezi, and Changji. The spread of HFM disease is also affected by wind speed. High wind speed brings unfavorable influence to the incidence in Khotan, Bayingholin, and Aksu, and has different results in Tacheng, Karamay, Bortala, Ili, and Shihezi. Let $\bar{\hat{\beta}}_j (j = 1, 2, 3)$ be the mean value of $\hat{\beta}_{jk}(u_{ik}, v_{ik}, t_k)$ at the $k$th times. Fig 7 visualizes the mean values of coefficient estimations in the GTWPR model. It reflects the relationship between HFM disease and these

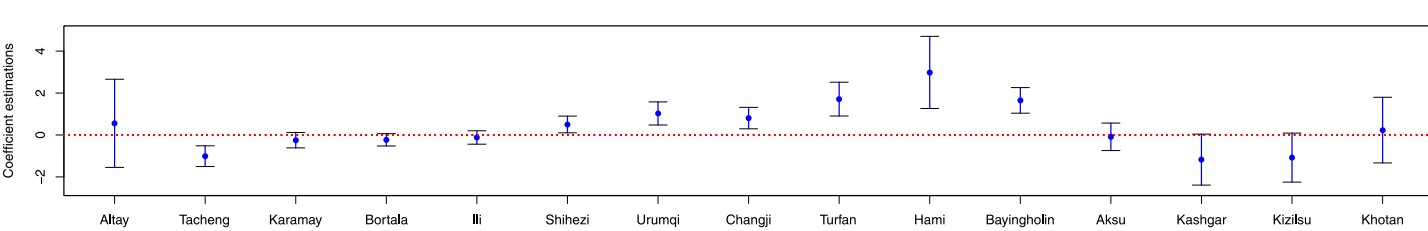

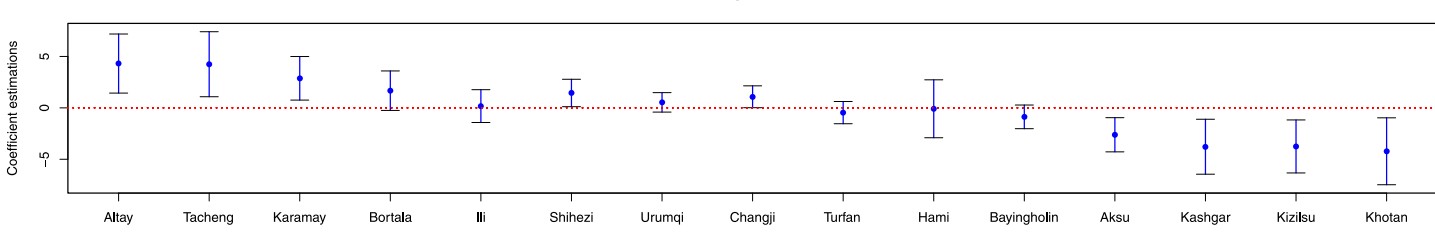

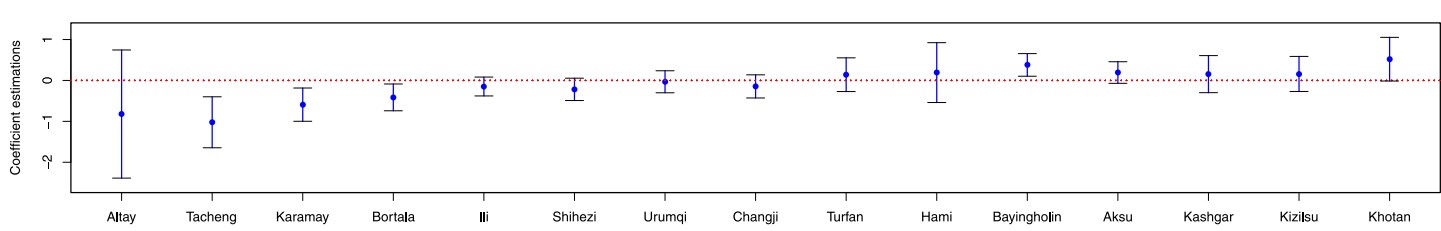

**Fig 6. The mean values and 95% confidence intervals of coefficient estimations.**

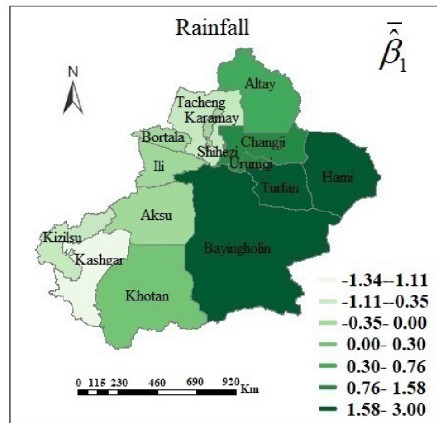
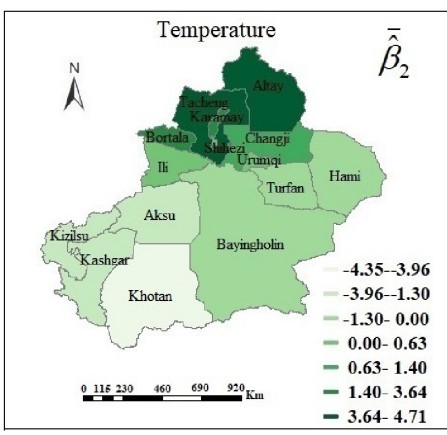
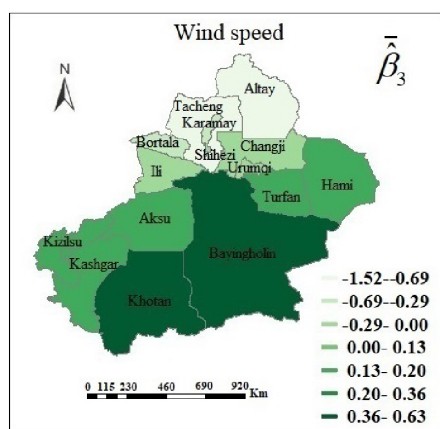

**Fig 7. The mean coefficient estimations of three meteorological factors.** It is generated by software ArcGIS 10.2 from http://eol.jsc.nasa.gov/SearchPhotos/.

three factors in various regions. The rainfall has a positive effect on HFM disease in Bayingholin, Turfan, and Hami. The cases in the northern have increased more quickly than in the southern with a growing process of temperature. However, the wind speed has a bigger affect on the southern than the northern.

## Socioeconomic factors

Cao et al. [28] obtained that the HFM disease was closely associated with population density and mobility. Huang et al. [12] showed that the influence of the tertiary industry and children was greater than that of the primary industry and middle school students. Thus, we select six socioeconomic factors to study their impact on HFM disease. In Xinjiang province, the gross domestic product (GDP) per capita is $42.40 \pm 86.51$(1000 yuan/people), the population density (PD) is $32.00 \pm 145.22$(10 people/km$^2$), the number of beds (NB) per thousand people is $58.83 \pm 46.01$(set/1000 people), the ratio of urban population (RUP) is $0.62 \pm 0.38\%$, the ratio of primary school (RPS) is $0.55 \pm 0.22\%$ and the ratio of tertiary industry (RTI) is $0.39 \pm 0.22\%$ in 15 regions from 2008 to 2017 (See Fig 8).

We first implement the principal component analysis to six socioeconomic factors. Load factors are selected as the explanatory variables when the cumulative proportion is more than 80%. The first three loads can be taken as the main parts of six socioeconomic factors in Table 5. Since the condition number of multiplex collinear test is $K = 1.143 < 15$ between the first three load factors, there is no significant correlation among them.

Let $z_{ijk}(j = 1, 2, \ldots, 6)$ be the six socioeconomic and $Z_{iJk}(J = 1, 2, 3)$ be three load factors at the $i$th ($i = 1, \ldots, 10$) region in the $k$th ($k = 1, \ldots, 15$) year, respectively. From Table 5, we have

$$
\begin{aligned}
Z_{i1k} &= 0.419z_{i1k} + 0.436z_{i2k} + 0.452z_{i3k} + 0.476z_{i4k} - 0.45z_{i5k}, \\
Z_{i2k} &= 0.347z_{i1k} - 0.414z_{i2k} - 0.349z_{i3k} + 0.248z_{i4k} - 0.137z_{i5k} - 0.711z_{i6k}, \qquad (5) \\
Z_{i3k} &= 0.573z_{i1k} + 0.381z_{i3k} - 0.176z_{i4k} + 0.698z_{i5k}.
\end{aligned}
$$

Suppose that $\vartheta_{ik}$ is the annual number of cases. The link function is $g(\vartheta_{ik}) = \eta_{ik} = \ln \vartheta_{ik}$. The GWPR model given by

$$
g_3(\vartheta_{ik}) = \alpha_0(u_{ik}, v_{ik}) + \alpha_1(u_{ik}, v_{ik})Z_{i1k} + \alpha_2(u_{ik}, v_{ik})Z_{i2k} + \alpha_3(u_{ik}, v_{ik})Z_{i3k} \qquad (6)
$$

for $i = 1, \ldots, 15$, where $k$ is a fixed constant taken from $\{1, 2, \ldots, 10\}$. The GTWPR model is

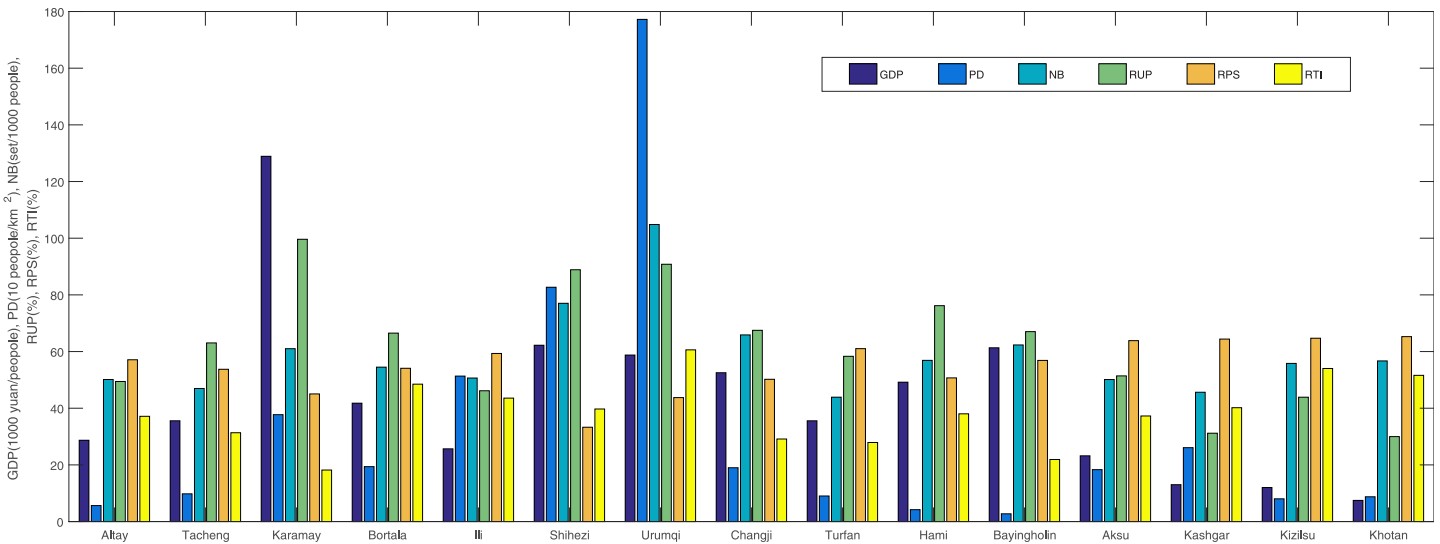

**Fig 8. Mean values of six socioeconomic factors in 15 regions of Xinjiang, 2008–2017.**

established as follows

$$g_4(\vartheta_{ik}) = \quad \alpha_{0k}(u_{ik}, v_{ik}, t_k) + \alpha_{1k}(u_{ik}, v_{ik}, t_k)Z_{i1k} + \alpha_{2k}(u_{ik}, v_{ik}, t_k)Z_{i2k}$$
$$+ \alpha_{3k}(u_{ik}, v_{ik}, t_k)Z_{i3k}, \quad k = 1, \ldots, 10,$$

(7)

where $\alpha_J(u_{ik}, v_{ik})$ and $\alpha_{Jk}(u_{ik}, v_{ik}, t_k)(J = 1, 2, 3)$ are the coefficients of three load factors at the $i$th region in the $k$th year. In Table 6, we compare the performance of two models. The larger the coefficient of determination $R^2$, the better fit model for the sample data. The values of AIC, D and MSE for GTWPR model are smaller than GWPR model. In view of this, GTWPR model are more suitable to investigate the influence of these three loads.

The estimated values $\hat{\alpha}_{Jk}$ of $\alpha_{Jk}$ can be obtained by the weighted least square method and local linear geographical weighted regression. The results are shown in Table 7. It reflects that the coefficient estimator has strong non-stationarity.

**Table 5. Loadings and cumulative proportion of the principal component analysis.**

| Number | Socioeconomic factors | I | II | III | IV | V | VI |
|---|---|---|---|---|---|---|---|
| 1 | GDP | 0.419 | 0.347 | 0.573 | 0.282 | 0.164 | 0.518 |
| 2 | PD | 0.436 | -0.414 | 0.000 | -0.104 | -0.784 | 0.259 |
| 3 | NB | 0.452 | -0.349 | 0.381 | 0.146 | 0.179 | -0.689 |
| 4 | RUP | 0.476 | 0.248 | -0.176 | -0.785 | 0.254 | 0.000 |
| 5 | RPS | -0.450 | -0.137 | 0.698 | -0.519 | -0.144 | 0.000 |
| 6 | RTI | 0.000 | -0.711 | 0.000 | 0.000 | 0.544 | 0.435 |
| Cumulative proportion | | 0.491 | 0.786 | 0.883 | 0.934 | 0.970 | 1.000 |

* I—The first load; II—The second load; III—The third load; IV—The fourth load; V—The fifth load; VI—The sixth load.

**Table 6. The comparison of models (6) and (7).**

| Model | $R^2$ | AIC | D | MSE |
|---|---|---|---|---|
| GWPR | 0.706 | 8222.77 | 8512.62 | 0.850 |
| GTWPR | 0.999 | 23.85 | 25.98 | 0.003 |

Substituting (5) into (7), we get the fitted model below

$$
\begin{aligned}
g_4(\hat{\vartheta}_{ik}) = \quad & \hat{\eta}_{ik} = \hat{\alpha}_{0k} + \hat{\alpha}_{1k}Z_{i1k} + \hat{\alpha}_{2k}Z_{i2k} + \hat{\alpha}_{3k}Z_{i3k} \\
= \quad & \hat{\alpha}_{0k} + (0.419\hat{\alpha}_{1k} + 0.347\hat{\alpha}_{2k} + 0.573\hat{\alpha}_{3k})z_{i1k} + (0.436\hat{\alpha}_{1k} - 0.414\hat{\alpha}_{2k})z_{i2k} \\
& + (0.452\hat{\alpha}_{1k} - 0.349\hat{\alpha}_{2k} + 0.381\hat{\alpha}_{3k})z_{i3k} + (0.476\hat{\alpha}_{1k} + 0.248\hat{\alpha}_{2k} - \\
& 0.176\hat{\alpha}_{3k})z_{i4k} - (0.45\hat{\alpha}_{1k} + 0.137\hat{\alpha}_{2k} - 0.698\hat{\alpha}_{3k})z_{i5k} - 0.711\hat{\alpha}_{2k}z_{i6k} \\
= \quad & \tilde{\alpha}_{0k} + \tilde{\alpha}_{1k}z_{i1k} + \tilde{\alpha}_{2k}z_{i2k} + \tilde{\alpha}_{3k}z_{i3k} + \tilde{\alpha}_{4k}z_{i4k} + \tilde{\alpha}_{5k}z_{i5k} + \tilde{\alpha}_{6k}z_{i6k}
\end{aligned}
\tag{8}
$$

for $i = 1, \ldots, 10$; $k = 1, \ldots, 15$; $j = 0, 1, \ldots, 6$, which can be considered as the estimated coefficients of six socioeconomic factors $z_{ijk}$. Let $\bar{\tilde{\alpha}}_j (j = 0, 1, \ldots, 6)$ be the mean value of $\tilde{\alpha}_{jk}$ at the $k$th times. Fig 9 describes the mean values and 95% confidence intervals of $\bar{\tilde{\alpha}}_j$. The influence of six socioeconomic factors also varies with the change of spatiotemporal locations. There exists significant spatiotemporal non-stationary in Kashgar, Kizilsu, Knotan, Altay, and Hami regions for the six factors. The mean values of coefficient estimations are given in Fig 10. In

**Table 7. Quantile and mean values of coefficient estimations and response variables.**

| Coefficient or variable | | Min | 1st Qu | Median | 3rd Qu | Max | Mean |
|---|---|---|---|---|---|---|---|
| True value | $\eta$ | 0.690 | 4.218 | 5.315 | 6.353 | 8.220 | 5.118 |
| Estimated value | $\hat{\alpha}_1$ | -153.590 | -2.835 | 0.110 | 3.436 | 71.863 | -2.835 |
| | $\hat{\alpha}_2$ | -12.473 | -0.321 | 1.954 | 4.204 | 50.137 | 3.596 |
| | $\hat{\alpha}_3$ | -18.669 | -2.097 | 0.410 | 4.759 | 397.309 | 9.358 |
| | $\hat{\eta}$ | 0.676 | 4.229 | 5.301 | 6.357 | 8.224 | 5.122 |

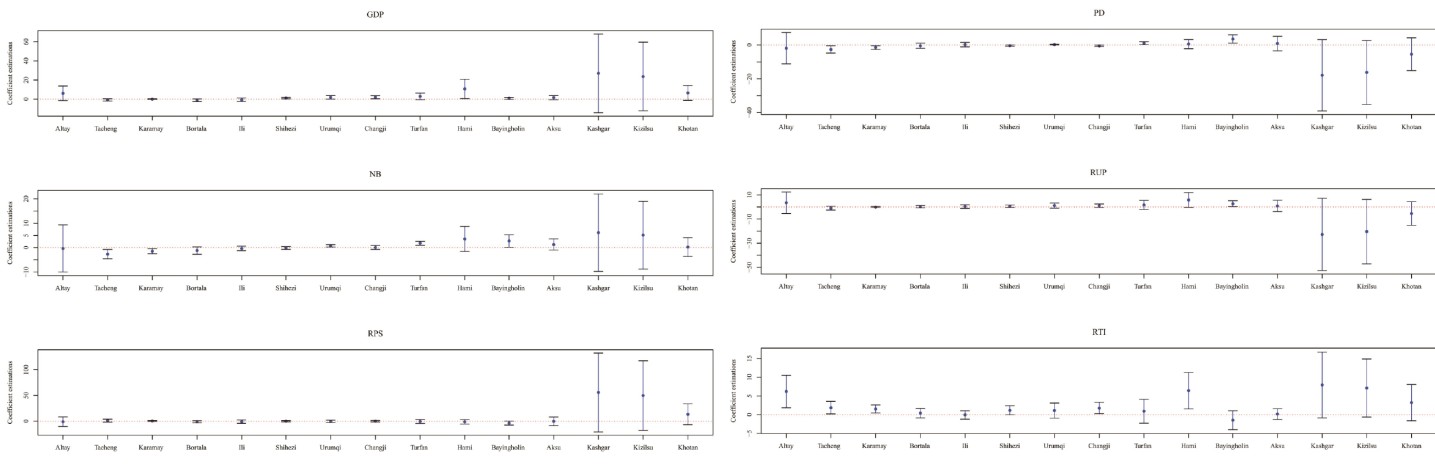

**Fig 9. The mean values and 95% confidence intervals of coefficient estimations.**

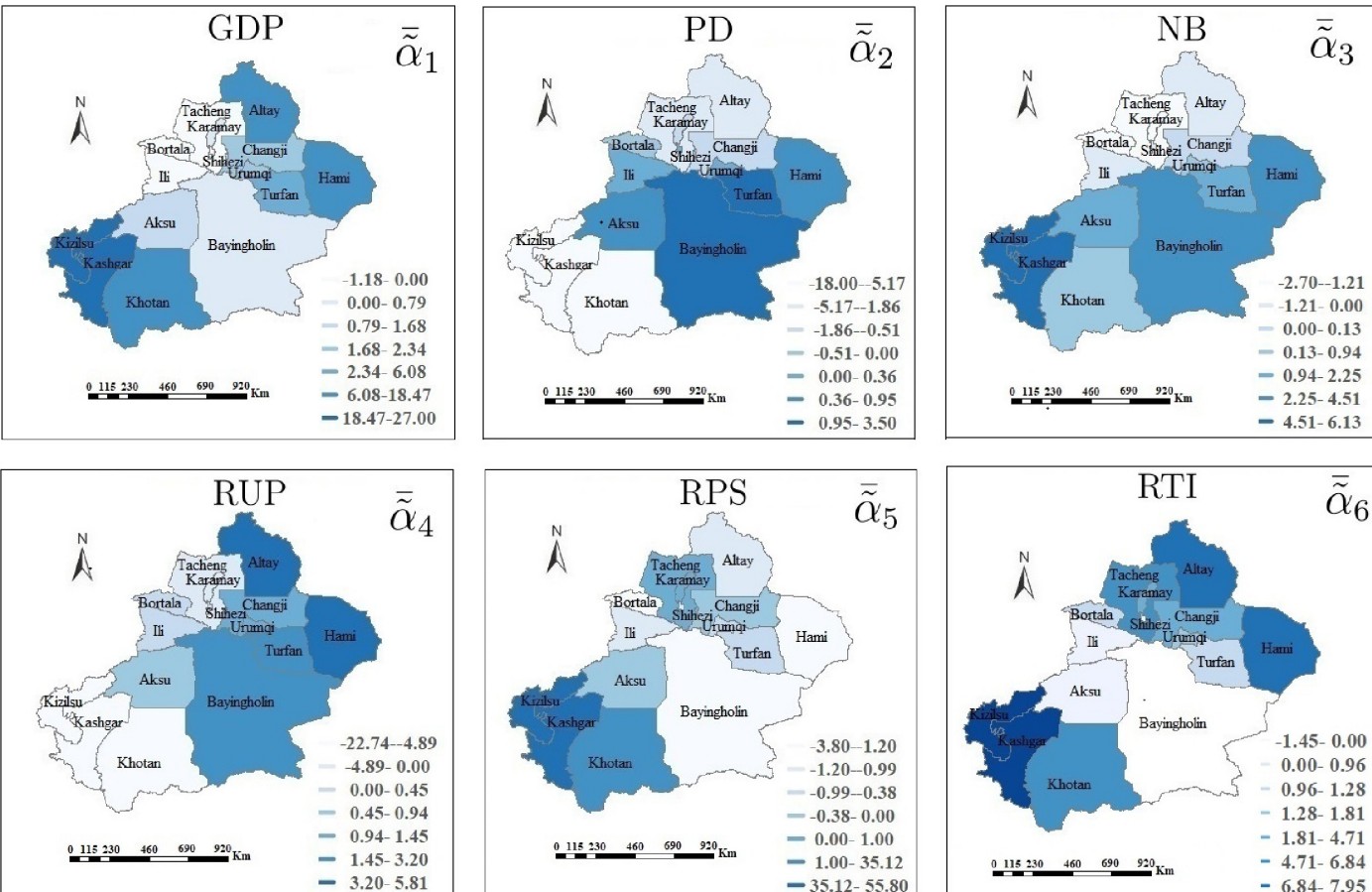

**Fig 10. The average coefficient estimations for six socioeconomic factors.** It is generated by software ArcGIS 10.2 from http://eol.jsc.nasa.gov/SearchPhotos/.

most regions, the number of HFM cases increases with the growth of GDP, RUP, and RTI, and reduction of NB. Moreover, PD always leads to the increase of the confirmed cases in Turfan and Bayingholin, and has different results in Tacheng, Karamay, Kashgar, Kizilsu, and Khotan. Higher RPS can give rise to lower HFM cases, such as Bayingholin. For the high-risk areas, Urumqi, Changji, and Shihezi are mainly affected by GDP, RUP and RTI. RTI, NB and PD are the main factors affecting HFM disease in Karamay and Tacheng. These six socioeconomic factors bring a smaller influence on Ili than other regions.

## Discussion

In this paper, we analyze the spatiotemporal characteristics and influencing factors of the HFM epidemic in Xinjiang from 2008 to 2017. In terms of the descriptive analysis, the number of male patients is more than that of females, which corresponds to the studies conducted in China [6], Northern Thailand [29], and other countries [1, 30]. Momoki [31] also found that boys were more susceptible to enterovirus infections than girls. In Xinjiang, the male-female ratio is 1.5:1, close to China's 1.56:1 from 2008 to 2009. Children aged 0–5 years have a higher risk of illness, accounting for more than 77% of the total cases. As revealed by the seroepide-miological analysis [32], more than 50% of children under the age of 5 lack the antibodies of HFM disease enterovirus. Zhang et al. [33] pointed out that Altay, Urumqi, Bayingholin,

Changji, and Ili had a high incidence in 2016. Due to the use of vaccine EV71, the number of cases had significantly decreased in 2017. Thus, vaccination is an effective measure for children to prevent HFM disease at present. Since the epidemic is contagious during the incubation period, early detection, treatment and isolation are necessary to prevent the spread of the epidemic.

Through the temporal distribution, there are one or two peaks for each year in Xinjiang, 2008–2017. The first peak usually exists in June, and the second smaller peak generally appears in October. Liu et al. [34] had a similar conclusion, in which the number of reported cases was mainly concentrated in May to July, accounting for 69.35% of all cases in Xinjiang from 2011 to 2015. Another smaller peak occurred in October to November. For geographical distribution, the incidence of the northern is higher than the southern from 2008 to 2017, mainly in Karamay, Urumqi, and Changji. The incidences are relatively small in Kashgar, Khotan and Kizilsu. This conclusion can be confirmed in [35].

Besides, the spatiotemporal scanning method is used to determine the cluster area and time of HFM disease in Xinjiang. Urumqi, Changji, Shihezi, Karamay and Tacheng always belong to the first cluster areas. The cluster time usually begins in April or May, and ends in August to November every year. Then, the spatiotemporal cluster of all cases is scanned from 2008 to 2017. The first cluster areas include Changji, Urumqi, Shihezi, Turfan and Karamay from April 2012 to September 2017. The cluster regions are mainly located in the northern.

Previous studies [11, 36] have shown that there exists a spatial non-stationary between this disease and influence factors. Different from them, the GTWPR model is proposed to investigate the spatiotemporal non-stationary, and analyze the relationship of meteorological and socioeconomic factors on HFM disease in Xinjiang from 2008 to 2017. It has a better fitting than the GWPR model by $R^2$, AIC, D and MSE. The result reveals that the influence of these factors has a spatiotemporal non-stationarity. Based on the meteorological factors, the HFM cases significantly increase in Turfan, Hami and Bayingholin, and decrease in Kashgar and Kizilsu as the rainfall is increased. With the temperature increasing, the HFM cases of the northern have increased more than that of the southern. However, the wind speed has a bigger impact on the southern than the northern. Kramer et al. [26] considered that the largest desert region and hotter summer were not beneficial for the survival of the virus. Moreover, Li et al. [8] indicated that the general association between wind speed with HFM disease might locally be weakened at high wind speed. Due to the unique terrain, the wind speed in the northern is much higher than in the southern. Although the increase of wind speed is conducive to the spread of the virus, they may stay in the air for a shorter time at a high wind speed [37].

The socioeconomic factors have significant spatiotemporal non-stationary on HFM disease. In most regions, the increase of GDP, RUP and RTI brings unfavorable influence to the disease. Furthermore, these three factors are important economic indicators for judging regional development. Huang et al. [12] indicated that the HFM cases were increased with the growth of GDP. Our results show that RUP and RTI also have similar effects. It reveals that the developed regions have a higher risk. NB is an indicator to describe the status of medical resources. The abundance of medical resources promotes the improvement of the health system, so that disease data can be recorded and reported in time [8]. Thus, the regions with high NB have more cases. The indicator RPS reflects the education level. The number of patients decreased with the increase of RPS. This means that people with higher education pay more attention to HFM disease. In addition, Cao et al. [28] pointed out that the higher PD made the virus easier to spread in urban areas. However, we find that the HFM epidemic has decreased with the increase of PD in Altay, Tacheng, Karamay, Kashgar, Kizilsu and Khotan. Therefore, spatiotemporal effects should not be ignored for controlling the HFM disease in the studied area.

## Conclusions

This article aims to investigate spatiotemporal features and influence factors of the HFM epidemic in Xinjiang from 2008 to 2017. The descriptive statistical analysis reveals some interesting facts. The male-female ratio is 1.5:1, among which the majority are children over 1 and under 5 years old. Generally, there exists an annual peak in June. The second smaller peak appears in October. The spread of HFM disease in the northern is more serious than that in the southern. The cluster area and time are determined by the spatiotemporal scanning method. It can reflect the high incidence areas and periods of each year. Urumqi, Changji, Shihezi, Karamay, and Tacheng belong to the first cluster regions every year. The clustering time mainly begins in April or May, and ends from August to November. The GTWPR model is established to analyze the relationship between this disease and influence factors. There exist significant spatiotemporal non-stationarity between HFM disease, meteorological and socioeconomic factors.

Further work also exists in our study. The spatiotemporal autocorrelation should be considered in the GTWPR model. Besides, other potential confounding variables have not been currently studied. It is worthy to be further studied.

## Supporting information

**S1 Data.**
(XLS)

## Author Contributions

**Methodology:** Shuman Sun.

**Supervision:** Zhiming Li, Xijian Hu, Ruifang Huang.

**Writing – original draft:** Shuman Sun.

**Writing – review & editing:** Zhiming Li.

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
