## [Decision Letter · Decision Letter 0]

23 Mar 2021

PONE-D-20-40568

Spatiotemporal characters and influence factors of hand, foot and mouth epidemic in Xinjiang, China

PLOS ONE

Dear Dr. Li,

Thank you for submitting your manuscript to PLOS ONE. After careful consideration, we feel that it has merit but does not fully meet PLOS ONE’s publication criteria as it currently stands. Therefore, we invite you to submit a revised version of the manuscript that addresses the points raised during the review process.

We look forward to receiving your revised manuscript.

Kind regards,

Tzai-Hung Wen, Ph.D.

Academic Editor

PLOS ONE

Journal Requirements:

'This work was supported by the Natural Science Foundation of Xinjiang Uygur Autonomous Region (XJEDU2017M001, 2018Q011), and the National Natural Science Foundation of China (U1703237, 11661076, 12061070).'

'The funders had no role in study design, data collection and analysis, decision to publish, or preparation of the manuscript.'

5. We note that Figures 1, 4, 7 and 9 in your submission contain map images which may be copyrighted.

You may seek permission from the original copyright holder of Figures 1, 4, 7 and 9 to publish the content specifically under the CC BY 4.0 license. 

If you are unable to obtain permission from the original copyright holder to publish these figures under the CC BY 4.0 license or if the copyright holder’s requirements are incompatible with the CC BY 4.0 license, please either i) remove the figure or ii) supply a replacement figure that complies with the CC BY 4.0 license. Please check copyright information on all replacement figures and update the figure caption with source information. If applicable, please specify in the figure caption text when a figure is similar but not identical to the original image and is therefore for illustrative purposes only.

Reviewers' comments:

Reviewer's Responses to Questions

5. Review Comments to the Author

Reviewer #1: The manuscript entitled “Spatiotemporal characters and influence factors of hand, foot and mouth epidemic in Xinjiang, China” applied multiple approach to analyze spatial and temporal patterns of HFMD in Xinjiang. Overall, the study design and analysis are sound and appropriate. Below are some comments:

Major issues:

1. What’s the main purpose to show the status of cases. It mainly reflects the distribution of age. It is a well-known knowledge that young children are the main target of HFM disease. May be the specific environmental condition of children staying at home or school in the study area could be discussed.

2. Line 213-215. The statement is confusing. Why do you ignore Kashgar, Kizilsu, and Khotan?

3. Line 231-233. What is the current strategy to block HFM transmission in the study area? Why increasing the frequency of children’s health examination can help to reduce transmission?

4. Lines 273-281. The climate condition can impact virus survival and transmission. How about socioeconomic factors? The statement only described positive or negative effects of these factors. How can you explain the associations?

Minor issues:

1. Line 16. The model you listed here are usually referred to “statistical models”.

2. Figure 4. How about the population distribution in the study area? Incidence rate could be used to compare the results.

Reviewer #2: Overall of the article scientifically sounds, however, several points need to be improved.

Abstract: The authors did not provide specific conclusion regarding the findings. The conclusion is too general and cannot reach to the point of usefulness or next steps. It would be better if the author can emphasize the significant of the findings for public health implication.

Introduction: The authors reviewed some articles and presented previous findings properly, however the authors did not provide rationale on the study area and justification to study all factors. Although the authors tried to present aims of the study between line 20-32, the issues did not present research hypothesis at all. Otherwise, this study is looked very alike a reference number 6 but just changed the study setting.

Also, statement such as section 2,3,4 and 5 did not relevant to the way of presentation in the followings part which use heading rather numbering.

Materials and Methods: The authors described source of data clearly but for meteorological data, the authors may consider to explain the data collection of the CMDC for example, satellite, meteorological station located at which area to provide clearly understanding how accurate the meteorological data for each area.

Since the author used spatial analysis, did the author apply spatial weight matrix?

Population structure in each area is different or similar which will affect the comparison of HFMD which strongly related with age. The authors should describe spatial context of each area in Xinjiang in particular from south to north, economic, living condition and so on.

Result: Table 1 will be the most benefit to the public health profession in Xinjiang, if the authors present specific rate rather than number of cases except variable" status". Can the downward trend of children attended school affect HFMD trend in the study area? Was a large outbreak happen in the year 2016? Can it affect spatio-temporal analysis?

Figure 3, for comparison among study area, the author may consider to apply standardized rate to present. As well as, the distribution of socioeconomic variables should be described. The author described principle component analysis to merge socioeconomic variables but it is not clear how the model selection was conducted. Since the narrative on page 14 line180-184 are not consistent with table 5.

The inverse association of meteorological data from north and south are interesting, is there any collinearity of the factors? or culture explanation?

Discussion: The author did not well explain the findings but rather compare with previous findings. It would be most interesting if the authors could explain the possible reasons why one factor affect differently to each area or even in opposite way.

4. Is the manuscript presented in an intelligible fashion and written in standard English?

Reviewer #2: No

---

## [Author Response · Author response to Decision Letter 0]

25 May 2021

1. Are these third party data (i.e., data not owned or collected by the author(s))?

No

2. If these are indeed third party data, please explain how others can access these datasets and confirm that others would be able to access these data in the same manner as the authors. Please also confirm that the authors did not have any special access privileges that others would not have.

No

3. If these are not third party data but there are ethical or legal restrictions on sharing a de-identified data set, please explain them in detail (e.g., data contain potentially identifying or sensitive information) and who has imposed them (e.g., a governmental body, an ethics committee, etc.). Please also provide contact information for a data access committee, ethics committee, or other institutional body to which data requests may be sent.

Yes, the data of HFM disease are obtained from Center for Disease Control and Prevention of Xinjiang Uygur Autonomous Region (http://www.xjcdc.com/). Authorization is required to access the database on this website. And the dataset should ont be used for any commercial purpose. We have provided the data as supporting information files. For some details of the data, please contact 1178843565@qq.com

4. If these are not third party data and there are no restrictions, please upload the minimal anonymized dataset necessary to replicate your study findings to a stable, public repository and provide us with the relevant URLs, DOIs, or accession numbers. For a list of recommended repositories, please see https://journals.plos.org/plosone/s/recommended-repositories. You also have the option of uploading the data as Supporting Information files, but we would recommend depositing data directly to a data repository if possible.

No

5. Please provide non-author contact information for a data access committee, ethics committee, or other institutional body to which data requests may be sent.

Yes, we have corrected it.

6. Additionally, please provide a URL for the Xinjiang Statistical Yearbook so interested readers can access the data in the same manner as the authors.

Yes, we have provide a website for the Xinjiang Statistical Yearbook (https://data.cnki.net/yearbook/Single/N2020040352).

---

## [Decision Letter · Decision Letter 1]

23 Jun 2021

Spatiotemporal characters and influence factors of hand, foot and mouth epidemic in Xinjiang, China

PONE-D-20-40568R1

Dear Dr. Li,

We’re pleased to inform you that your manuscript has been judged scientifically suitable for publication and will be formally accepted for publication once it meets all outstanding technical requirements.

Kind regards,

Tzai-Hung Wen, Ph.D.

Academic Editor

PLOS ONE

Additional Editor Comments (optional):

Reviewers' comments:

Reviewer's Responses to Questions

**Comments to the Author**

1. If the authors have adequately addressed your comments raised in a previous round of review and you feel that this manuscript is now acceptable for publication, you may indicate that here to bypass the “Comments to the Author” section, enter your conflict of interest statement in the “Confidential to Editor” section, and submit your "Accept" recommendation.

Reviewer #1: All comments have been addressed

Reviewer #2: All comments have been addressed

2. Is the manuscript technically sound, and do the data support the conclusions?

Reviewer #1: Yes

Reviewer #2: Yes

3. Has the statistical analysis been performed appropriately and rigorously? 

Reviewer #1: Yes

Reviewer #2: Yes

4. Have the authors made all data underlying the findings in their manuscript fully available?

Reviewer #1: Yes

Reviewer #2: Yes

5. Is the manuscript presented in an intelligible fashion and written in standard English?

Reviewer #1: Yes

Reviewer #2: Yes

6. Review Comments to the Author

Reviewer #1: All the comments have been addressed by the author. The current version manuscript acceptable for the journal.

Reviewer #2: Although the authors did not explain or respond to my comments on the response to the reviewer(s). I found some revision according to my comments.

7. PLOS authors have the option to publish the peer review history of their article (what does this mean?). If published, this will include your full peer review and any attached files.

Reviewer #1: No

Reviewer #2: No

---

## [Editor Report · Acceptance letter]

7 Jul 2021

PONE-D-20-40568R1 

Spatiotemporal characters and influence factors of hand, foot and mouth epidemic in Xinjiang, China 

Dear Dr. Li:

I'm pleased to inform you that your manuscript has been deemed suitable for publication in PLOS ONE. Congratulations! Your manuscript is now with our production department. 

Kind regards, 

on behalf of

Dr. Tzai-Hung Wen 

Academic Editor

PLOS ONE